# Peridot: Accelerating Out-of-Core GCN Data Reuse Pattern and Co-Design on GPU

Shakya Jayakody*, Jun Wang*

*University of Central Florida.

*shakya@ucf.edu, jun.wang@ucf.edu*

*Abstract*—**Graph Convolutional Networks (GCNs) are pivotal in a diverse array of applications, including scientific research, engineering, biomedical protein-protein interactions (PPI), and natural language processing (NLP). The demand for efficient GCN computation has catalyzed extensive research into GPU acceleration techniques. However, a persistent challenge in this domain is managing out-of-core data, which exceeds the storage capacity of limited GPU memory, resulting in significant data movement latency and underutilization of GPU computational resources.**

**This paper introduces Peridot, an innovative framework that leverages NVIDIA GPUDirect Storage (GDS) technology coupled with sophisticated memory allocation strategies to enhance the performance of out-of-core GCNs on GPUs. Peridot is engineered to significantly reduce data movement latency by orchestrating efficient transfers of sparse matrix data between the GPU and system memory, particularly during sparse chain matrix multiplication. It optimizes memory usage within the GPU to support larger matrices than previously possible. The system incorporates a dynamic memory allocation scheme tailored to the sparsity patterns of matrices, reducing unnecessary memory consumption and improving data locality. Additionally, by utilizing GDS technology, Peridot enables direct, high-speed data transfers between storage devices and GPU memory, bypassing the CPU and reducing the overhead associated with traditional data transfer methods. Our evaluations demonstrate that Peridot substantially surpasses baseline models, offering considerably low latency in both synthetic and real-world graph benchmarks. These improvements are particularly notable in scenarios involving extensive GCN data, where inefficiencies in data movement and memory allocation have historically been significant obstacles.**

## I. INTRODUCTION

Graph neural networks (GNNs) have emerged as a powerful tool for processing structured data, particularly in the field of natural language processing (NLP), and biomedical applications. One of the most critical operations in GNNs is the sparse matrix multiplication (SpGEMM) for modeling graph structures [7], [9], [18]. However, the computational demands of this operation can be formidable, especially when dealing with large-scale graphs, motivating the need for efficient acceleration techniques [4], [7], [19]. The formidable computational demands of SpGEMM, which are central to GNNs, stem largely from the enormous size and inherent sparsity of matrices typical in applications such as biomedical data modeling. For instance, the Protein-Protein Interaction (PPI) networks, which may encompass approximately 68 million nodes and 1.5 billion edges, exemplify such challenges [15], [21]. The adjacency matrix for a graph of this scale would need to accommodate roughly 68 million x 68 million entries. Even

assuming a sparse format with only 1% of these entries being non-zero, this configuration still results in about 680 million active entries that require significant computational resources to manage [7], [9].

The advent of high-performance computing and the ever-increasing demand for processing large datasets have propelled the development of more efficient computational methods and hardware accelerators [12]. Among these, GPUs have emerged as a pivotal technology, offering substantial parallel processing capabilities far beyond traditional CPUs. This advancement is particularly beneficial in the realm of sparse matrix operations, which are foundational to GCN [2], [10]. Sparse matrices, characterized by a majority of zero-valued elements, pose unique challenges and opportunities for optimization GCN performance [1]. Traditional dense matrix operations are computationally and memory intensive, making them impractical for sparse matrices, especially when dealing with large-scale data [13]. Two common formats for representing sparse matrices are the CSR and CSC formats [3], [17].

This paper introduces Peridot, a novel approach to GPU memory out-of-core GCN acceleration that harnesses the power of GPUs. We focus on accelerating the SpGEMM of matrices represented in GCN, Leveraging the low-latency data movement between the hierarchical memory layers of GPU memory, DRAM, and NVMe enhances the overall efficiency of GCN. Our motivation for Peridot stems from the need for an efficient, scalable, and adaptable solution for GCN out-of-core computation on GPUs. Our approach focuses on optimizing memory access patterns, reducing the latency, and maximizing parallel execution.

Our hardware-software co-design contributions can be summarized as follows:

- Automatic Data Compression Technique: This technique utilizes operator hints to detect sparsity based on a tiered memory hierarchy and selects the appropriate sparse matrix compression algorithms for matrices A and B, respectively.
- Automatic Data Transfer Technique: This method selects a low-latency, zero-copy path through a GPU-directed storage approach. It enables direct loading of sparse matrices to the GPU, bypassing the CPU to minimize CPU involvement. Additionally, verify whether the CPU is used to compress the matrix before transfer.
- Metadata and Data Decoupling Technique: This approach is designed for out-of-core SpGEMM computations to

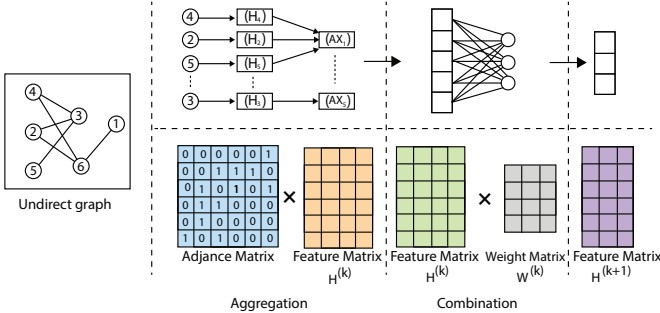

Fig. 1. Chain Matrix Multiplication in a Graph Convolutional Network Layer During the Aggregation and Combination Phases

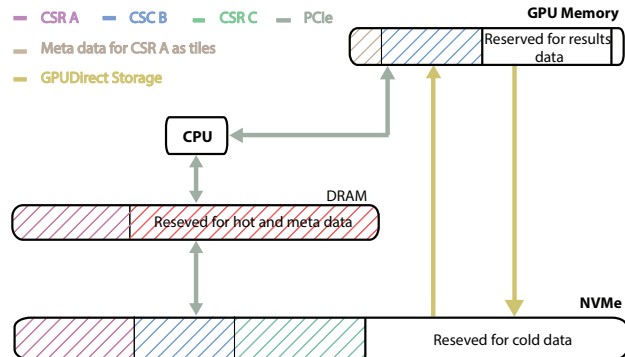

Fig. 2. Peridot memory utilization

optimize space allocation within a tiered memory system, including High Bandwidth Memory (HBM), GDS, and host DRAM.

## II. BACKGROUND AND MOTIVATION

### A. Graph Convolutional Networks

GCNs exhibit parallels to traditional convolutions in the realm of graph analysis, particularly in their approach of sharing "filter" parameters across all graph locations. However, a distinguishing characteristic of GCNs is their dependence on message-passing techniques. This implies that vertices within a graph engage in information exchange with their neighbors via the transmission of "messages". To express GCNs in more mathematical terms, we first need to determine how to aggregate the various messages received by a node. Given the varying number of messages across nodes, an operation that can handle any number of inputs is required. Common approaches include summation or averaging. Let's define the previous features of nodes as $H^{(k)}$. Then, a GCN layer can be expressed as:

$$H^{(k+1)} = \sigma \left( \hat{D}^{-\frac{1}{2}} \hat{A} \hat{D}^{-\frac{1}{2}} H^{(k)} W^{(k)} \right)$$

Here, $W^{(k)}$ are the weight parameters that transform the input features into messages, i.e., $H^{(k)}W^{(k)}$. We modify the adjacency matrix $A$ by adding the identity matrix $I$ to it, resulting in $\hat{A} = A + I$. This modification ensures that each node includes its own features in the message-passing process.

To average the messages rather than summing them, we compute the matrix $\hat{D}$, a diagonal matrix where each diagonal element $\hat{D}_{ii}$ denotes the number of neighbors (including the node itself) node $i$ has the symbol $\sigma$ denotes an activation function, which is not limited to the sigmoid function. In practice, ReLU-based activation functions are often used in GNNs. Fig. 1 the computation in a GCN layer during the aggregation and combination phases.

The aggregation equation is defined as:

$$\text{Aggregate}^{(k)} = \hat{D}^{-\frac{1}{2}} \hat{A} \hat{D}^{-\frac{1}{2}} H^{(k)} \tag{1}$$

where $\hat{A}$ is the augmented adjacency matrix with self-connections, and $\hat{D}$ is the diagonal degree matrix of $\hat{A}$.

The combination equation with the transformation through weight parameters and an activation function is given by:

$$H^{(k+1)} = \sigma \left( \text{Aggregate}^{(k)} W^{(k)} \right) \tag{2}$$

where $W^{(k)}$ represents the weight matrix at layer $k$ and $\sigma$ is the activation function, such as ReLU.

### III. PERIDOT CO-DESIGN FOR OUT-OF-CORE GCN

GCNs, integral to processing relational data, employ adjacency and weight matrices that benefit substantially from compression into CSR and CSC formats. Our focus is on streamlining the data architecture to facilitate efficient processing during various computational phases. Specifically, during the aggregation phase, the adjacency matrix is handled in CSR format, designated as CSR A, while the feature matrix is managed in CSC format, termed CSC B. This configuration is inverted during the combination phase to maximize computational efficiency. Fig. 2 illustrates the memory utilization of Peridot during the aggregation and combination phases.

i. **Automatic Data Compression Technique:** This technique applies advanced algorithms to compress matrices into CSR and CSC formats, facilitating more efficient data processing and storage. By utilizing operator hints, the system intelligently detects sparsity within the data based on a tiered memory hierarchy, selecting the most effective compression algorithms for each matrix. This not only optimizes memory utilization but also enhances the speed and scalability of GCN operations.

ii. **Automatic Data Transfer Technique:** To further reduce latency and streamline data flow, this technique employs a GPU-directed storage approach, enabling a low-latency, zero-copy path for the direct loading of sparse matrices to the GPU. This method minimizes CPU involvement and verifies the necessity of CPU-based matrix compression prior to data transfer. By optimizing the data pathway, this technique ensures that crucial computational resources are preserved for processing rather than data handling.

iii. **Metadata and Data Decoupling Technique** Designed for out-of-core SpGEMM computations, this strategy optimizes space allocation within a tiered memory system—including High Bandwidth Memory (HBM), GPUDirect

Storage (GDS), and host DRAM. By decoupling metadata from actual data, the system can more effectively manage memory allocation, enhancing both the efficiency and speed of data access during the computation phases.

- **Aggregation Phase:** Initiating with an analysis of the adjacency matrix's byte size in CSR format and the feature matrix in CSC format, we assess the data's compatibility with available GPU and DRAM memory capacities, avoiding actual data loading to minimize initial overhead. This assessment guides the deployment of GPU Direct Storage to directly transfer the compressed feature matrix (CSC B). Following this, CSR A's metadata is transferred to DRAM, with strategic portions cached into GPU memory as tiles to ensure optimal data availability for subsequent computations. This preparation is crucial for handling dynamically required data, such as the updated feature matrix $H^{(k+1)}$, and facilitates seamless data conversions.
- **Combination Phase:** In the post-aggregation state, the CSR formatted feature matrix $H^{(k+1)}$ is relocated to DRAM, and the subsequent layer's weights $W^{(k+1)}$ in CSC format are prepared within the GPU. This configuration ensures that all critical data is optimally placed within the GPU's memory constraints. After computations, the updated feature matrix $H^{(k+1)}$ can archived back to NVMe storage, supporting error correction or future computations. This phase is critical for reducing latency and maximizing bandwidth, enhancing the overall computational throughput of the GCN.

Our approach systematically addresses the conventional bottlenecks of memory limitations and data transfer delays inherent in GCNs. Preliminary results indicate substantial improvements in both latency reduction and bandwidth utilization, thus advancing the efficiency of GCN operations.

## IV. Evaluation

### A. Methodology

Peridot integrates sparsity-aware optimizations that specifically reduce the I/O transfer overhead associated with zero-copy operations. The architecture of Peridot strategically employs the CSR format for matrices on the left-hand side of multiplications and the CSC format for the right-hand side. This deliberate choice maximizes memory coalescence and minimizes memory access latency on the GPU, crucially enhancing the efficiency of sparse matrix multiplications. These multiplications often suffer from irregular memory access patterns and require careful management to make efficient use of GPU memory bandwidth.

The performance of Peridot was evaluated using an undirected graph suite of sparse matrices from the SuiteSparse matrix collection detailed in Table I. Which provides a diverse set of real-world sparse matrices. The evaluation metrics focused on speedup factors compared to baseline.

The experimental setup for evaluating the Peridot is meticulously designed to ensure a robust and comprehensive assessment of its performance in GCN. This setup encompasses the hardware configuration, software environment, and the selection of datasets for testing. NVIDIA RTX 4090, chosen for its high computational capability, extensive memory, and support for CUDA, making it ideal for high-performance computing tasks. CUDA Toolkit Version 12.2 for Peridot enabling GDS and baseline both using Cuda.

In our experiment, we evaluate Baseline, Peridot, and Peridot without GDS:

- *Baseline*: The baseline configuration involves storing 50% of both the adjacency matrix and the feature matrix directly in GPU memory, with the remainder stored in DRAM. This setup reflects a conventional approach where data is statically partitioned between GPU memory and system memory without dynamic reallocation based on computational demands.
- *Peridot*: Pre-analysis of the graph's adjacency and weight matrices. It employs an automatic data transfer technique that dynamically allocates the optimal proportion of the adjacency matrix and associated data for GPU and system memory storage. Peridot utilizes GPU Direct Storage (GDS) to facilitate efficient and rapid data transfers back and forth between the GPU and system storage, optimizing computational throughput and reducing memory transfer latency.
- *Peridot without GDS*: In contrast to its GDS-enabled counterpart, Peridot without GDS also performs a preliminary analysis of the adjacency and weight matrices but relies solely on traditional memory transfer methods. It uses the same automatic data transfer technique to determine the optimal data allocation for the adjacency matrix but confines data transfer activities to DRAM and CPU interactions.

### B. Experimental Results and Analyses

The performance comparison table Table II clearly delineates the capabilities of the Peridot system both with and without GPU Direct Storage (GDS) against a traditional baseline. This analysis focuses on two key metrics: bandwidth and latency, which are critical for assessing the efficiency of data-handling techniques in GCNs.

The results demonstrate that Peridot significantly enhances bandwidth across all datasets when compared to the baseline configuration. For instance, in the kmer_V2a dataset, the bandwidth increases from 3952.42 MB/s in the baseline to 6566.6 MB/s with Peridot employing GDS, marking a substantial improvement. This indicates that the integration of GDS effectively optimizes data transfer processes, allowing for faster handling of large-scale graph data.

Even without GDS, Peridot shows improved bandwidth over the baseline, although the gains are less pronounced compared to the GDS-enabled configuration. This suggests that the core optimizations in Peridot, such as advanced memory management and data transfer techniques, contribute positively to performance but are significantly enhanced by leveraging GDS technology.

## TABLE I
MEMORY REQUIREMENTS AND GPU MEMORY CONSTRAINTS FOR SUITESPARSE GRAPH DATASETS [5]

| Dataset | No. Vertices | No. Edges | Memory Req. (GB) | GPU Mem. Restrict. (GB) |
|---|---|---|---|---|
| kmer_V2a | 55.04M | 117.21M | 4.63 | 4 |
| kmer_U1a | 67.71M | 138.77M | 5.52 | 4 |
| mycielskian18 | 196.6K | 300.93M | 9.63 | 8 |
| kmer_P1a | 139.35M | 297.82M | 11.76 | 10 |
| kmer_A2a | 170.72M | 360.58M | 14.27 | 12 |
| kmer_V1r | 214M | 465.41M | 18.317 | 16 |

## TABLE II
PERFORMANCE COMPARISON OF GCN DATA HANDLING TECHNIQUES

| Dataset | Baseline | | Peridot | | Peridot without GDS | |
|---|---|---|---|---|---|---|
| | Bandwith (MB/s) | Latency (ms) | Bandwith (MB/s) | Latency (ms) | Bandwith (MB/s) | Latency (ms) |
| kmer_V2a | 3952.42 | 1054.15 | 6566.6 | 529 | 4761.79 | 729.502 |
| kmer_U1a | Out of core | – | Out of core | – | Out of core | – |
| mycielskian18 | 3514.33 | 2285.49 | 7104.39 | 1017 | 4739.62 | 1524.42 |
| kmer_P1a | 3504.27 | 3773.47 | 7307.5 | 1207 | 4782.98 | 1844.07 |
| kmer_A2a | 3660.48 | 4526.35 | 7355.86 | 1455 | 4893.4 | 2187.19 |
| kmer_V1r | 3745.9 | 5104.49 | 7132.88 | 1926 | 4820.03 | 2850.17 |

The latency metrics further reinforce the benefits of the Peridot system. With the integration of GDS, latency is markedly reduced across all datasets. For example, the latency for the mycielskian18 dataset decreases from 2285.49 milliseconds in the baseline to 1017 milliseconds with Peridot using GDS, more than halving the delay involved in processing. This reduction is crucial for applications requiring real-time processing of graph data, such as dynamic network analysis and interactive data visualization.

Peridot without GDS also demonstrates reduced latency compared to the baseline, although not as dramatically as its GDS-enabled counterpart. This indicates that while GDS plays a significant role in optimizing latency, the inherent design of Peridot itself is geared towards more efficient data processing.

## V. RELATED WORK

**A Systematic Survey of General Sparse Matrix-Matrix Multiplication** This survey meticulously categorizes and evaluates a wide range of algorithms and optimizations developed for SpGEMM, offering insights into their applicability across various computing platforms, including CPUs, GPUs, and distributed systems. The authors provide a detailed classification based on the algorithmic strategies employed for SpGEMM, such as row-wise, column-wise, and hybrid approaches, alongside discussing the merits and limitations of each method. They also explore the evolution of parallel computing techniques that enhance SpGEMM performance on modern high-performance computing architectures. Special attention is given to optimizations that address the challenges posed by the sparse nature of the matrices, such as minimizing memory access latency and maximizing computational efficiency [6].

**Sextans: A Streaming Accelerator for General-Purpose Sparse-Matrix Dense-Matrix Multiplication** introduce a novel hardware accelerator designed to efficiently handle the sparse-matrix dense-matrix multiplication (SpMM), a fundamental operation in various scientific and data analytics applications. The Sextans accelerator is specifically tailored to enhance the performance of SpMM operations by leveraging a streaming architecture that optimizes data flow and minimizes memory access latency. The Sextans framework distinguishes itself by adopting a general-purpose approach that accommodates a wide range of sparse matrix formats and densities, ensuring broad applicability across different domains [14].

**Accelerating sparse matrix–matrix multiplication with GPU Tensor Cores** their methodology encompasses a detailed analysis of the memory access patterns and computational strategies that can be optimized for sparse matrices, ensuring that the Tensor Cores are utilized efficiently despite the inherent challenges posed by irregular data structures [8], [20]. This research work not only contributes to the computational mathematics and high-performance computing fields by demonstrating the feasibility and benefits of adapting sparse computations to Tensor Cores but also sets a foundational framework for future explorations into optimizing other sparse linear algebra operations using emerging GPU features [11], [16].

## VI. CONCLUSION

Peridot is an innovative approach designed to significantly enhance the performance of out-of-core GCNs through the strategic use of GDS. Our results demonstrate that Peridot, by integrating advanced data transfer techniques and memory management strategies, offers substantial improvements in processing large-scale graph data, as evidenced by marked increases in bandwidth and reductions in latency across multiple datasets. This optimization is particularly evident in scenarios where the GDS technology is employed, showcasing its capability to handle intensive data transfers addressing the critical challenges of memory limitations and data transfer

delays that have historically impeded the scalability of graph processing applications.

## Acknowledgements

This work was sponsored in part by the U.S. National Science Foundation (NSF) under Grants 1907765, and 2400014.

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
