# OpenReview forum: "Peridot: Accelerating Out-of-Core GCN Data Reuse Pattern and Co-Design on GPU"
_iscaconf.org/ISCA/2024/Workshop/MLArchSys — MLArchSys 2024 OralPoster_

### Official Review · Reviewer_arip · 2024-05-21
**Hierarchical memory management for GNN training**

**Confidence:** 4
**Rating:** 6

**Detailed Feedback And Questions For Authors:**

Summary:
This paper addresses training large GNNs, which require offloading certain matrices “out-of-core” (i.e., off of GPU HBM) in order to train the model given limited GPU memory. The authors propose Peridot, a framework which leverages hierarchical memory (GPU HBM, CPU DRAM, and NVMe) to manage sparse feature matrices. Peridot schedules and manages the movement of these matrices during the aggregation and combination phases of training across memories to enable training larger GNNs while minimizing the overheads of data movement between GPU and offloaded memory. Furthermore, Peridot leverages data compression techniques for sparse matrices.

Detailed comments
- As mentioned above, while Peridot seems like an effective system, the presentation in Section III could benefit from additional details to understand Peridot’s technical contribution. For example, what operator hints passed to Peridot? What compression algorithms are evaluated - and how do hints help in evaluation? How does decoupling metadata allow for better memory allocation?
- I understand a part of the limited detail is due to the short page limit of workshop papers. However, at the same time, I feel like the related work discussion (Section V) is quite long and can be significantly shortened, especially given that it only discusses 3 papers. I would much rather see this space used for more details about Peridot.
- I appreciate the evaluation, which shows that Peridot is still effective without GDS. However, it would be useful to better understand how Peridot leverages the characteristics/trade-offs across each memory tier. Why not just use GDS or DRAM entirely? Perhaps more detail on Peridot’s memory management mechanism as mentioned above would help clarify this point (e.g., what is “hot data” in Figure 2).
- Table II seems to have a typo - mycielskian18 reports 24739 MB/s for Peridot without GDS, which seems inconsistent with the other results.

**Top Reasons To Accept The Paper:**

- Well-designed system that leverages co-designed hierarchical memory and GPUDirect storage alongside software compression/sparsity to improve GNN training throughput.
- Strong preliminary results to demonstrate the effectiveness of the proposed system.

**Top Reasons To Reject The Paper:**

- Limited details on the specific components of Peridot (“operator hints”, “effective compression algorithms”, etc.) makes it difficult to understand its research contributions and potential.
- While Peridot’s techniques do improve performance over the baseline, much of its benefit appears to from a straightforward application of GPU Direct storage.

---

### Official Review · Reviewer_vHoR · 2024-05-24
**The paper introduces Peridot, a framework that improves the performance of out-of-core GCNs on GPUs using NVIDIA GPUDirect Storage and advanced memory allocation strategies, but lacks detailed explanations and quantitative analyses of key techniques, such as automatic data compression and memory management decisions.**

**Confidence:** 4
**Rating:** 5

**Detailed Feedback And Questions For Authors:**

This paper introduces Peridot, an framework designed to enhance the performance of out-of-core Graph Convolutional Networks (GCNs) on GPUs by leveraging NVIDIA GPUDirect Storage (GDS) technology and advanced memory allocation strategies. The key challenges targeted by this paper include managing out-of-core data that exceeds GPU memory capacity, which results in significant data movement latency and underutilization of GPU computational resources. Peridot tackles these challenges by orchestrating efficient transfers of sparse matrix data between GPU and system memory during sparse chain matrix multiplication, optimizing memory usage to support larger matrices, and utilizing GDS technology for direct, high-speed data transfers between storage devices and GPU memory. The system incorporates a dynamic memory allocation scheme tailored to matrix sparsity patterns, reducing unnecessary memory consumption and improving data locality.

- I understand that this is a short 4-page paper illustrating the preliminary achievements towards a significant direction of research. However, I believe in some cases the paper could provide a bit more detail or more clearly explain the concepts. For instance, for the automatic data compression techniques, why is it “automatic”? What are the “advanced” algorithms used to compress matrices into CSR and CSC formats? How do they facilitate more efficient data processing and storage? Another example is the decisions made about hot/cold data and handling the CPU memory and GPU direct accesses.

- A quantitative execution analysis along with Figure 2 could be very helpful to understand the contribution and impact of each category of data transfer.

- The evaluations could be improved by further expanded discussions making clear connections between the observations and the three key contributions of the paper.

**Top Reasons To Accept The Paper:**

Peridot enhances the performance of out-of-core GCN computations on GPUs by leveraging NVIDIA GPUDirect Storage and advanced memory allocation strategies, demonstrating improvements in latency and efficiency.

**Top Reasons To Reject The Paper:**

The paper lacks detailed explanations and quantitative analyses of key concepts and techniques, such as the specifics of automatic data compression algorithms and the decisions regarding memory management, limiting the clarity and comprehensiveness of its contributions.

---

### Official Review · Reviewer_mN3c · 2024-05-27
**This work is a good start but not ready for acceptance.**

**Confidence:** 4
**Rating:** 4

**Detailed Feedback And Questions For Authors:**

The authors are presenting their co-design for GCN data reuse.  While I believe this is a good topic to attack, I am not sure this paper has reached a point where it is ready for acceptance at this workshop.  The issues range from the novelty of the work to some issues I have with the writing.

The first issue I have for this work is the novelty feels limited.  The first item is the separation of the metadata and the coefficients but that has been done before.  For example, in ESCALATE: Boosting the Efficiency of Sparse CNN Accelerator with Kernel Decomposition by Li et al. they talk about masks for sparsity.  It can also be seen when using adjacency matrices for graph computations.  I appreciate the authors' work on implementing the solution and using GPU direct storage, I feel this work may be incremental at best.  The idea of GPU direct storage is to hold values in a high bandwidth storage off chip and so using it for this application does not seem very novel.  This issue may be due to the lack of fine details about the approach and implementation.  This brings me to the second issue, which is the lack of details.  I was expecting some more hardware in the solution and more information on the overall implementation.  This was lacking in the submitted version of the paper in my opinion.

The final issue I would like to see addressed by the authors as they move forward with their work is the writing.  In particular, the related works section should highlight similarities and differences between previous works and the current work under consideration.  The authors did not appear to do this in this case.  That weakened the argument and lowered value of the submitted paper.

**Top Reasons To Accept The Paper:**

The acceleration of large models is important and GCNs are a good topic.
The work uses the current GPU direct storage to evaluate so it is using modern mechanisms.

**Top Reasons To Reject The Paper:**

I am not sure of the novelty of this work.
There is a lack of details on the techniques that prevents a deep understanding of the implementation and mechanisms.
There are some writing improvements I would like to see

---

### Official Review · Reviewer_jMSD · 2024-05-28
**Peridot: Accelerating Out-of-Core GCN Data Reuse Pattern and Co-Design on GPU**

**Confidence:** 4
**Rating:** 6

**Detailed Feedback And Questions For Authors:**

Thank you for submitting the paper to MLArchSys. The problem identified in the paper is timely and utilizing GPUDirectStorage to bypass the CPU is necessary especially in large scale deployments especially in datacenters where CPU can become the bottleneck, reducing the benefits from GPU acceleration, as per Amdhal's law.

I feel the paper can be improved by providing better details as to how the different proposed optimization techniques are implemented. Such as providing clearer explanations for what these “advanced algorithms” are etc.

I also have a few other questions which the authors can ponder/address for future full paper submission.

1. How does the system determine the most effective compression algorithms for each matrix?
2. What is the role of the CPU and GPU in the automatic data transfer technique? It was not clear who performs what.
3. How does the metadata and data decoupling technique manage memory allocation?
4. How does the system ensure that all critical data is optimally placed and what is "optimal" within the GPU’s memory constraints during the combination phase?
5. While the results show that GDS is faster than accessing the CPU memory, I assume the GPU/CPU have unified memory in state-of-the-art systems, therefore I belive reading directly from the unified memory should be faster than GDS in terms of latency.

**Top Reasons To Accept The Paper:**

1. The proposed idea implements a system that address key bottlenecks in GCN computations, especially for large-scale graphs which are emerging today.

2. The use of GPU-directed storage to directly load sparse matrices onto the GPU is a good optimization and necessary since we need to bypass CPU, which has been the traditional method for all data orchestration. It improves data transfer efficiency.

3. The experimental setup and evaluation are comprehensive.

**Top Reasons To Reject The Paper:**

The paper has some weaknesses that if addressed could provide more clarity on several technical aspects.

1. For instance, the specific term "advanced algorithms" used for compression and the term "operator hints" are not clear. In fact, the body of the paper does not provide much detail into how the system is implemented.

2. Figure 2, which is supposed to explain the memory utilization process during the aggregation and combination phases, is not clearly explained thus making it confusing as to what really the tiering and data movement optimization are.

3. It is also unclear whether the GPU or CPU performs the data compression/decompression.

---

### Decision · Program_Chairs · 2024-05-30

**Decision:**

Accept (Oral/Poster)

**Comment:**

Congratulations! We are pleased to inform you that your paper has been accepted for presentation at MLArchSys 2024. We look forward to your participation at the workshop. Further details regarding the schedule and format will be provided soon. See you at the workshop!